# UNILIPID, a Methodology for Energetically Accurate Prediction of Protein Insertion into Implicit Membranes of Arbitrary Shape

**DOI:** 10.3390/membranes13030362

**Published:** 2023-03-21

**Authors:** André Lanrezac, Marc Baaden

**Affiliations:** Laboratoire de Biochimie Théorique, CNRS, Université Paris Cité, 13 rue Pierre et Marie, Curie, F-75005 Paris, France

**Keywords:** implicit membrane, lipid bilayer insertion, hydrophobicity scale, coarse-grained representation

## Abstract

The insertion of proteins into membranes is crucial for understanding their function in many biological processes. In this work, we present UNILIPID, a universal implicit lipid-protein description as a methodology for dealing with implicit membranes. UNILIPID is independent of the scale of representation and can be applied at the level of all atoms, coarse-grained particles down to the level of a single bead per amino acid. We provide example implementations for these scales and demonstrate the versatility of our approach by accurately reflecting the free energy of transfer for each amino acid. In addition to single membranes, we describe the analytical implementation of double membranes and show that UNILIPID is well suited for modeling at multiple scales. We generalize to membranes of arbitrary shape. With UNILIPID, we provide a methodological framework for a simple and general parameterization tuned to reproduce a selected reference hydrophobicity scale. The software we provide along with the methodological description is optimized for specific user features such as real-time response, live visual analysis, and virtual reality experiences.

## 1. Introduction

Proteins perform various biological functions in their role as essential components of cells. An important role of proteins is that they can be part of the cell membrane and contribute to the architecture and function of these fluid envelopes that protect the cell from external influences [1]. The positioning of a protein in the bilayer is an important factor in many regulatory circuits. It can be predicted computationally, while direct determination by experiment is often impractical [2]. Now and then, direct correspondence of computational prognoses with experimental determinations is conceivable. For example, NMR data such as NOE signals observed between aromatic protons and nondeuterated lipid chains can be combined with computational studies. Such a combination was used to determine the position of a peptide in a membrane environment using an interactive experiment [3]. Electron paramagnetic resonance (EPR) is an alternative experimental approach that provides important information about protein insertion [4]. Computational methods therefore play a crucial role in the study of protein insertion into membranes. Basic computer models range from simple implicit representations of the membranes involved [5] to computationally demanding simulations of fully hydrated lipid bilayers using molecular dynamics approaches [6,7]. Such computational approaches are often calibrated to reproduce experimental measurements performed to evaluate the free transfer energies of amino acids into the membrane environment. The computational methods can then in turn be used to predict the free transfer energies in more complex situations under a variety of conditions. Hydrophobicity scales are useful tools in this context because they can be used to predict the hydrophobicity or hydrophilicity of amino acids, which is a measure of their affinity for water or oil. Simply put, these scales reflect the fact that hydrophobic amino acids tend to prefer a nonpolar environment and are often found buried inside proteins, while hydrophilic amino acids tend to prefer a polar environment and are often found on the surface of proteins or in loop regions. When a hydrophobic amino acid is exposed on the surface of a protein, it contributes to its ability to insert into the membrane. Therefore, the hydrophobicity of an amino acid and its accessibility to the surface can significantly affect the behavior of a protein within a membrane. Several scales of hydrophobicity have been developed that assign a numerical value to each amino acid based on its observed behavior in a given context. These values can be used to predict the behavior of an amino acid in a particular environment or to compare the hydrophobicity of different amino acids. Common hydrophobicity scales include the Kyte-Doolittle scale [8], the Hopp-Woods scale [9], the Eisenberg scale [10], and the Fauchère-Pliska scale [11]. These scales are useful for predicting the behavior of amino acids within a membrane and for understanding the function of membrane proteins.

There are some limitations to the use of hydrophobicity scales to predict membrane protein insertion. In their design, hydrophobicity scales do not take into account the secondary or tertiary structure of proteins, which is a drawback of the methodology. If the global structure of a protein is known, a surface map of hydrophobicity could be a useful tool for incorporating the protein into the membrane, and there are methods that move in this direction [12,13,14]. In this case, a specific energy scale would need to be derived to serve as the basis for an implicit membrane model.

Here we will build on our implementation of the Integrative Membrane Protein and Lipid Association Method (IMPALA) for interactive simulations [15], which is based on the Fauchère-Pliska scale. Our goal is to extend the previous methodology to achieve energetically accurate prediction of protein insertion into implicit membranes of arbitrary shape. This goal is directly related to the development of a software package optimized for specific user functions such as real-time responses, live visual analysis, and virtual reality experiences. The IMPALA reference technique originated from Brasseur’s research team to define the relationship between different proteins and a membrane structure. We chose this approach because it is simple and computationally efficient enough to lend itself to live calculations with user interaction and possible inclusion in an integrative modeling process. Early studies in the literature using an original implementation of the IMPALA method helped characterize synthetic new cell-penetrating peptides [16] or known peptides [17]. Other studies addressed the transformations of a nisin compound in the bilayer. Such studies also confirmed experimental results [18]. The alignment and insertion depth of integral membrane proteins were furthermore studied in [19]. Inspired by the capabilities of the Protein Positioning in Membranes (PPM) approach for predicting membrane insertion, we sought to improve our interactive IMPALA prototype. The latest version of PPM, PPM 3.0, enables prediction of protein arrangements in a variety of membrane environments, including planar, curved, and multiple membranes. While PPM uses an anisotropic solvent model to make these predictions, our goal is to maintain the simplicity of an implicit technique while overcoming the downsides of the original IMPALA model.

For this very reason, we will only briefly examine the setup of molecular dynamics simulations, another common application for membrane insertion, for which a variety of tools exist. In the article by Javanainen and Martinez-Seara, many available tools are presented [20]. These tools that help design complex explicit membranes include the Insane tool [21], CHARMM-GUI [22,23], some more specific approaches such as polyply to easily create models for polymers [24], or the ability to create large and highly curved systems using TS2CG [25]. Our focus, however, is on interactive membrane insertion simulations, which are too costly with an explicit membrane representation. Implicit membrane models provide a well-balanced sweet spot for such real-time approaches. The goal of this work is to make a practical contribution by providing an extended methodology for dealing with implicit membranes. While the main contribution is not in theory, we clarify and improve important aspects not previously discussed in the literature, and provide a methodological framework for a simple and general parameterization tuned to reproduce the selected reference hydrophobicity scale.

The ability to interactively study membrane insertion of proteins in such implicit models is critical to our own work on membrane fusion, and we believe that the approach described in this work provides a unique and powerful method for this purpose. In our earlier work, we published an updated implementation of the implicit membrane model originally developed by Brasseur’s group. The focus of this earlier work was to ensure the correctness of the implementation and to enable interactive real-time simulations. However, upon further investigation, we discovered several important limitations of the model, which we address in this work with a new and generalized methodology. We realized that these limitations were actually easy to overcome, which led us to develop an extended implicit model called UNILIPID, which stands for **UN**iversal **I**mplicit **LI**pid-**P**rote**I**n **D**escription. This model borrows from the original model developed by Brasseur, but goes far beyond its properties. One of the most important new features of UNILIPID in this regard is its ability to accurately reproduce the targeted hydrophobicity scale and to allow us to easily re-parameterize to change the level of representation, e.g., between all atoms and different degrees of coarse-grained simplification. The second aspect that is important to us is the prediction of protein insertion into implicit membranes of any shape. In particular, we will focus on membrane models that have more complicated shapes than just a planar bilayer, as is often the case in membrane fusion. In this work, we present the UNILIPID model and show that it is capable of reproducing lipid-protein association energies.

More specifically, in this paper we address an important limitation of the original Brasseur model that arises from the decision to use only 7 unique atom types for all amino acids. The resulting drawbacks can be clearly seen in Figure 3 from the paper by Ducarme [26] which shows Brasseur’s original graph where significant deviations from the reference data for GLN, LYS, MET, SER, and TRP occur. Our goal is to obtain a transfer energy for each amino acid that exactly matches the reference scale. To this end, we have extended the model to use optimised atom types for the amino acids, hence increasing the number of overall atom type variants. With this extension, we are able to perfectly reproduce the reference scale free transfer energies for each amino acid. We have generalized the model to easily parameterize the atomic type parameters for any reference transfer energy data. Our model is designed to fit any experimental scale, e.g., the Fauchère-Pliska scale and many others. The approach is intrinsically independent of the representation scale and the method can be applied to all scales, from all-atom models to coarse-grained models such as Martini [27] to other models with multiple beads such as OPEP [28] or for a single bead as in the Levitt model [29]. We compare the original Martini scale for the free transfer energy with a reparameterization using the Fauchère-Pliska scale. To facilitate the use of our approach, we provide reference implementations for all-atom, Martini, and Levitt-level representations. In addition to extending the implicit membrane representation, we aim to account for lipid-specific membrane thickness and implement double membrane systems. These systems are important in biology, for example in membrane fusion, and our implementation allows for programmatic movement of the two membrane planes. This functionality can be achieved analytically by modifying the simple 1D function C(z) used in IMPALA. Of course, if we abandon the idea of analytic formulas to describe membrane shape, we can generalize to arbitrary surface shapes and membrane shapes by using the 3D distance to a triangulated mesh surface instead of a unidimensional representation of C(z). With this mesh-based implementation, we can immediately implement double-membrane scenarios or even more complex scenarios.

With these contributions of the present work, we now have a solid methodology for performing parameterizations for such interactive experiments for a chosen energy reference scale and with membranes of arbitrary shape.

## 2. Materials and Methods

We first explain the design of the parameter set and the calibration strategy we implemented, and refer to the refinement algorithms we used. We then describe the extension to other types of membrane compositions and the implementation of an analytical double membrane model. This approach is then generalized to membranes of arbitrary shapes. Limitations are pointed out.

### 2.1. Parameter Set Design and Calibration

In the original work on the IMPALA method [26], the authors decided to use 7 atom types to represent all amino acids: Csp3 for single bonded carbons, Csp2 for double bonded and aromatic carbons, H(=0) for uncharged hydrogen (bonded to C, further referred to as Hnc), H(/0) for charged hydrogen (further referred to as Hc), O for oxygen, N for nitrogen, and S for sulfur. For each of these atom types, a transfer energy per accessible surface area was parameterized to reproduce the experimental transfer energy scale by Fauchère and Pliska [11] used as a reference. Since these atom types were chosen to be universal for all amino acids, their parameterization resulted in a compromise where the transfer energy for certain amino acids is very close to the experimental one (less than 2% error: Arg, Asp, Cys, Gly, Phe, Tyr, Val), reasonably close (less than 50% error: Glu, His, Ile, Leu, Thr, Trp), or quite off (more than 50% error: Ala, Asn, Gln, Lys, Met, Ser). The value for proline was not reported in the original publication. We have recalculated it and it is off by 21%. Here we propose to relax this design decision by reparameterizing the atom types per residue, as close as possible to the original parameters but as different as necessary to obtain transfer energies for each of the amino acids that are very close to the experimental values. Alternatively, one may use reparametrization to reproduce any hydrophobicity reference scale.

In practise, we implemented four methods to iteratively adjust the initial parameters of the original atom types until the targeted experimental reference energy E_tr/EXP_ is reproduced. For this purpose, standard configurations for each amino acid were used to calculate the transfer energy based on the original IMPALA approach summarised in Equations (1)–(4), for details see [15]. The predicted transfer energy corresponds to the value of E_IMP_ summed over the side chain atoms of the amino acid under study when it is located in the middle of the bilayer, where z = 0.
C(z) = 0.5 − (1 + exp(α(|z| − z_0_)))^−1^,(1)
E_int_ = −∑_i = 1_^N^ S_(i)_ E_tr(i)_ C_(zi)_,(2)
E_lip_ = a_lip_ ∑_i = 1_^N^ S_(i)_ C_(zi)_,(3)
E_IMP_ = E_int_ + E_lip_,(4)

In these equations, the value of α defining the width of the lipid-headgroup interface is 1.99—tuned to reproduce a headgroup region of 4.5 Å width–, z_0_ is the position of the centre of the lipid headgroup region in the model fixed at 15.75 Å, S_(i)_ is the surface area of atom i exposed to the solvent, E_tr(i)_ is the parameterized transfer energy per surface area for the atom type of atom i, a_lip_ is −0.018 kcal mol^−1^ Å ^−2^ and i runs over the indices of all side chain atoms of the residue under study. It is the values of E_tr(i)_ that are then iteratively adjusted until the resulting E_IMP_ matches E_tr/EXP_.

There are many ways to adjust the initial parameters to reproduce target values. Here we have used four simple approaches that we developed for implementing UNILIPID. Our overall goal was to minimise deviations from the initial atom type parameterization as long as the target energy is calculated correctly. Therefore, the parameterizations that provide the correct energies are not unique. For our purposes, we selected the parameterization from the four approaches for each amino acid that required the fewest changes compared to the initial set. The detailed description of these methods for calibrating atomic types have been moved to the Appendix A, so the core of the manuscript focuses more on the methodology and less on the specific implementation of the calibration approaches. This includes Appendix A describing pseudo-code for calibration method 4.

### 2.2. Extending the Lipid Bilayer Structure Representation

The original parametrization of Milik and Skolnick [30], on which IMPALA is based, corresponds to a theoretical bilayer model with fixed geometry and a total width of 36 Å, two polar headgroup regions with a width β of 4.5 Å, and a central layer consisting of the acyl chains with a width of 27 Å. These parameters can be related to the experimentally measurable hydrocarbon thickness 2D_C_ and steric thickness D_B_’ as defined in references [31,32] with the following equations:β = (D_B_’ − 2D_C_)/2,(5)
z_0_ = (D_B_’ − β)/2,(6)
α = 2 ln(88.0145)/β,(7)

The IMPALA parameters correspond fairly closely to the structural properties of DOPC, as shown in Table 1. We derive six additional parameter sets for other lipids from the literature, spanning bilayer widths up to 47.8 Å. Other parameter sets for higher or lower temperatures and for additional phospholipids such as PS, PI, SM, and cholesterol could be derived if coherent structural reference parameters are obtained.

These parameters only mimic the structural properties of these bilayers, but have no effect on the insertion energetics expressed by the parameters E_tr(i)_ and a_lip_. Such an energetic reparameterization would require knowledge of a consistent set of lipid-specific transfer energy measurements, to which we currently do not have access. It should be noted, however, that such reparameterization is entirely possible. Since we do not currently have such parameters, we have not implemented such a feature. Should such data become available in the future, UNILIPID could easily be adapted to mimic membranes composed of multiple lipid species. In this case of mixed lipid bilayers composed of different species, specific terms of E_tr(i)_, a_lip_, α, and z_0_ would be assigned to each lipid species for each atom type. Thus, for a given particle, we could perform a random draw at each fixed time step to determine which type of lipid the particle interacts with. The probability of encountering each lipid type would be directly related to its proportion in the membrane or could even be spatially distributed among the voxels of the simulation box.

### 2.3. Extending UNILIPID to Double Membranes

In the original model, the membrane is analytically represented by the C(z) function given in Equation (1). This equation can be adjusted to analytically implement a double membrane environment C_double_(z) as follows:C_double_(z) = C_main_(z) + C_side_(z) × C_side_(−z),(8)
C_main_(z) = 0.5 − (1 + exp(α(|z| − (m + z_0_))))^−1^,(9)
C_side_(z) = 1 − (1 + exp(α((z + m) − z_0_)))^−1^,(10)

The analytically implemented double membrane C_double_ consists of a main equation for C_main_, representing a single membrane whose thickness can be varied by adding a term m to z_0_ (Equation (9)). The combination of two symmetrically opposed half-membranes on z = 0, which retains this thickness term, is given by C_side_ (Equation (10)). For z + m, a shift of the equation along z is possible, which serves to represent the free internal interfaces of the double membrane system for all m > 18, when they merge for m ∈ [13.5, 18], and after fusion ∀ m ∈ [0, 13.5]. Note the lack of an absolute value for z to produce the representation of a half-membrane, and a modified constant so that C_side_ is shifted by 0.5, and the resulting term C_side_(z) × C_side_(−z) ∈ ± 0.5. These scenarios are depicted in Figure 1.

The analytic implementation makes it easy to dynamically move the membranes apart or closer together, either interactively or programmatically. We implemented this possibility in our interactive implicit membrane simulation using the UnityMol [34] and BioSpring [35] programmes, as described in detail in [15], by connecting a slider to adjust the value of m. In the visualisation, the primitives representing both membranes are shifted to follow the variation of m. In this double membrane representation, the inner layers have textures with a transparency value that follows C_double_(z), as shown in Figure 2.

### 2.4. Generalization to Arbitrary, and Multiple, Membrane Shapes

To further generalise the approach presented so far and extend it to arbitrary membrane shapes, we propose to define the desired membrane by a triangulated mesh of the surface of its centre plane. This mesh is then used in the original implementation instead of a reference plane. Multiple membranes are simply represented by multiple meshes. Let us describe this approach in detail.

#### 2.4.1. Representing Membranes via Meshes

For our purpose, we decided to generate some reference mesh structures programmatically. First, the size of the membrane mesh is defined by its length and width in Å. It represents the centre of the membrane, i.e., each vertex of this surface, whether flat or arbitrarily shaped, has a coordinate corresponding to a zero value on the *z*-axis of the C(z) function, indicating its position with respect to the implicit membrane environment. This reference surface is a vertex grid in which a resolution variable is defined in Å, corresponding to the distance between two rows or two columns. Figure 3 shows a representation of this surface.

We have xSize rows and ySize columns, so a total of (xSize + 1) (ySize + 1) vertices. The mesh is triangulated to be able to graph the surface. Each square is represented by two triangles, and each triangle is then listed by the three indices of the vertex that makes it up. These indices are arranged in a unique order so that the normals of the triangles all point to the same side of the surface. This process results in a total number of triangles of (xSize × ySize) × 2, each with 3 vertices. We then create a copy of this surface with reversed normals to represent the mesh for the adjacent bilayer leaflet and the two 4.5 Å thick layers that form the interface of the implicit membrane. In the UnityMol software, it is possible to show or hide these latter layers, as it can sometimes be interesting not to show them in order to observe certain aspects of the insertion in more detail (Figure 4).

Another possibility is to vary the curvature of the membrane, where curvature is defined as the reciprocal of the radius of a sphere. In the procedure where we loop over all vertices, we can apply any mathematical formula to the z (vertical) component of the vertex that is a function of the x (width) and y (length) components. For the curvature, for example, we obtain in this way the equation for a sphere: $z=\pm \sqrt{r^{2}-x^{2}-y^{2}}-r$.

Rather than explicitly defining the membrane surface through an equation (Figure 5), the user can provide a custom mesh that can be designed with third-party software or possibly derived from experimental data such as image stacks by segmentation [36,37]. Required mesh data includes the list of vertices, triangles, and normals. This data is sent from UnityMol, which acts as a graphical user interface, to the BioSpring simulation engine to calculate the corresponding UNILIPID potential. This data is transparently exchanged via the MDDriver middleware.

Note that in developing such a software implementation, we focus in particular on interactive simulations. Such simulations are very special in many respects, and because of their particular requirements, otherwise commonly used visualization software cannot easily be used for interactive simulations. For simplicity, we provide a general video tutorial in the data associated to the manuscript to more clearly demonstrate the use case of this interactive membrane insertion and to guide the user through the use of the software.

#### 2.4.2. Using Meshes as Reference for Implicit Membrane Lipid Association Calculations

To replace the original implicit membrane calculations that assumed a planar membrane and implement an implicit membrane based on a particular mesh shape, the position of each particle i in this new reference frame must be recalculated, as shown in Figure 6.

For these calculations, the mesh points are first read into BioSpring as a point cloud. This point cloud is then sorted for efficient distance determination using hash functions. We are currently still improving the performance of this part. Then, for each particle, we determine the closest distance to the point cloud along the normals to the surface. The normals are used to judge which side of the membrane a particular particle is on and to assign a sign to z. This information is needed to calculate the insertion depth of the protein. In practice, we determine the vector between the center of mass of the protein and the nearest vertex. Then we calculate the angle in the range of −90 and 90 between this vector and the vector normal to this vertex. Based on the positive or negative angle, we can determine on which side the center of mass of the reference surface is located and estimate the insertion depth. To determine the corresponding insertion angle, we subtract from the positions of the two particles defining the insertion vector the two vectors of these particles with respect to their nearest vertices.

### 2.5. Flexible Elastic Network Representation of the Protein

Some interactive experiments require flexibility of the protein. An efficient way to achieve this effect is via elastic network models. We have previously described the methodoloy in [35,38,39]. We have implemented such features in the BioSpring software, which uses additive energy and force terms to represent the various contributions to a given model representation. BioSpring features an extended elastic network model (aENM) that combines the spring network with non-bonded terms between atoms or pseudoatoms. The aENM model supports the use of multiple cut-off distances to determine a multi-component spring network. In this study, a single cut-off distance of 9 Å without non-bonded terms was used.

## 3. Results

We present the results obtained with UNILIPID. First, we compare our new parameterization with previous parameterizations and the experimental transfer energy scale of Fauchère and Pliska [11], with which the original method was calibrated. We then calibrate for different hydrophobicity scales and consider the results. We investigate the effect of the parameters in our method on the observed membrane insertion behavior and consider implementations at different scales of representation. We conclude with selected use cases to demonstrate the potential of the UNILIPID approach on a double membrane system and on a membrane of arbitrary shape.

### 3.1. Comparison of Previous IMPALA Implementation with UNILIPID

We introduced four methods to fit the initial parameters from the implementation of IMPALA [15] to the chosen experimental transfer energy scale, in this case that of Fauchère and Pliska [11]. We generalized the algorithm by systematically testing the 4 methods for calibrating each amino acid and then retaining only the results of the method whose energy differences between the calibrated atom types and the initial values are minimal. Therefore, we obtain a mixture of the results from the different methods in the final set of parameters. The final fitted values can be found in Appendix A. These values can be quickly adjusted using our approach for any representation type, from all atoms to coarse-grained levels. Two types of input are needed for this purpose: a database of PDB structure files to derive the surface information per particle, and a correspondence file linking particle types to their reference transfer energies. The following Figure 7 shows the comparison of the transfer energy for each side chain for the experimental scale with the original IMPALA implementation and with the UNILIPID reparameterization. It is adapted from the data in the original Figure 3 in the article by Ducarme [26].

From this figure, it can be seen that all side chains now reflect the reference energy scale with high accuracy. In particular, amino acids such as lysine, for which the wrong sign of the transfer energy was originally found, now reproduce the expected physicochemical properties. Figure 8 shows in detail the changes made for each atom type during parameter calibration.

The above figure shows how much the energy values for each atomic type of these new amino acid parameterizations differ from those of the original ones. In (D), the deviations, which appear very high in percentage terms, must be related to the fact that some starting values in the original implementation are far off the target energies. It is apparent that little adjustments are generally needed for the Csp2, N and O atom types. The S and Hc atom types are only significantly modified for one or two amino acids: Met and Cys for sulfur, and Thr and Leu for Hc. Csp3 and Hnc are more frequently adjusted in the procedure.

### 3.2. Implementing Several Hydrophobicity Scales

By construction, the approach presented here is easily extendable to any hydrophobicity scale. For illustration, we implement two scales calibrated in the recently published coarse-grained Martini 3 force field for membrane systems [40]. Overall, the procedure is as follows. The necessary information can be found in the Appendix A on pages 15 to 17. The free transfer energies were extracted from this work and used as target values for our method. Here we use the calculated theoretical surface areas of each grain type (normal, small, and tiny) to obtain an equivalent for the transfer energy per unit surface area of each atom in kJ mol^−1^ Å^−2^, as shown in Table 1 of [26]. We use the scales for transfer to hexadecane and to octanol to calibrate two Martini 3 parameter sets for our experiments in UNILIPID.

In order to adapt the Martini 3 parameters, it was necessary to adapt the algorithm for managing the residue representation—and thus the particles. We used the database of conformations mentioned above, from which all amino acids are then extracted. The tool *martinize* [41] is used to generate structures and topologies. The structures are used to calculate for each amino acid the surface area of the grains they form in the Martini representation, using the tool FreeSASA [42]. The topologies are used to define for each amino acid the types of Martini 3 grains and associate them with their transfer energy, which is contained in “APPENDIX 3” of the summary document “Summary_v3.0.b.3.2.pdf” accessible in version 3.0.b.3.2 of the MARTINI 3.0 CG Force Field (open beta) at URL http://cgmartini.nl/index.php/martini3beta (accessed on 7 February 2023).

Additional adjustments were made as follows. First, two adjustments were made using the distribution values of hexadecane (HD -> WN) and octanol (OCOS -> WN). Then, similar fits were made to two more of these values multiplied by the theorethical surface area of the associated grain to obtain energy values in kJ mol^−1^ Å^−2^ as in [26]. The fitting method was chosen to preserve the relative ratios between the energy values of the grain types of each amino acid. The main reason for this is that, unlike the all-atom scale fits, we do not have values to refer to that are on the same order of magnitude as the values associated with the calculation of the transfer energy in IMPALA. Therefore, the fit in terms of amplitude can be quite high. We considered it necessary to keep the proportions between the grain types in order to preserve the physicochemical meaning determined by the parameters in the molecular dynamics simulations that served as a reference for Martini 3, since these simulations were validated by reproducing experimental distribution data [27].

### 3.3. Effect of Mesh Parameters on the Quality of Results

An interesting question is what is the optimal mesh resolution to obtain accurate results while optimizing performance by using the coarsest possible mesh spacing. Let us clarify that we are referring here to the numerical (in)accuracy that results from coarsening the mesh. Our goal is to find the optimal tradeoff between speed and accuracy in our simulation. This evaluation is independent of the reference chosen and applies to any hydrophobicity scale at a given degree of mesh coarsening. By definition, the “ideal” reference value is the one where no mesh is used, which in practice is very close to a fine mesh spacing (below the Angstrom scale). Since we specifically target interactive methods, the coarser the mesh chosen, the faster and smoother the simulation.

Here, we investigate the sensitivity of UNILIP to this parameter to determine the extent to which the coarseness of the surface affects the accuracy of the membrane insertion results in terms of insertion depth and insertion angle. To this end, we test a planar membrane sheet tessellated with various degrees of roughness. We then perform membrane insertion of the OmpA membrane protein, which served as the main test case throughout the method development. The coarser the grid, the more the putative normal vector between the protein particle and the membrane is distorted compared to the ground truth normal vector, leading to errors in the forces acting on the system during insertion.

Figure 9 illustrates the interactive insertion of OmpA and summarizes the results of a single interactive experiment in which the analytical membrane representation is successively changed to the mesh representation at 10 resolutions. It specifically compares data such as time/angle/depth/frame for the original (“ideal”) recalibrated IMPALA implementation in UNILIPID that still uses an analytical membrane with the mesh-based approach where the mesh grid spacing is decreased from 10 to 1 Å to progressively increase resolution. The results are from a single simulation session of 6 min of interactive insertion, without human intervention, with stabilization of about 30s for each grid spacing parameter set. The experiment was performed with both UnityMol and BioSpring running on the same computer. All calculations were performed on multiple threads using OpenMP parallelization of our BioSpring software on an Intel^®^ Core™ i5-10500 @ 3.1 GHz (released in 2020)-6 cores, using 8 threads. The nearest vertex search algorithm was not optimized. The OmpA all-atom representation contains 2586 particles for the whole system. The number of grid points that needs to be probed for each resolution in the order [10→1] are [100, 100, 144, 169, 225, 324, 484, 784, 1764, 6724].

Comparison of the analytical membrane representation with a mesh representation for the OmpA membrane protein shows that the insertion depth stabilizes at about −6 Å. This value -which varies considerably- is found approximately for a resolution of 4 Å, as indicated by a black arrow. This grid spacing provides a sweet spot between the speed and fluidity of the interactive run and the loss of accuracy compared to a reference method without mesh discretization. We note that the value for the insertion angle of about 60° is easier to obtain. This angle value is already quite close for a resolution of 4 Å (highlighted by a black arrow). In these interactive experiments, we measured the number of iterations per second of the BioSpring program. The update rate decreases with roughness, i.e., with increasing resolution. At a resolution of 4 Å, it is about ⅓ of the rate of the reference analytical implementation. To illustrate our setup, we show an example of the protein simulated in a planar mesh with the reference insertion depth at z = 0 Å, a size of 90 × 90 Å, and a resolution of 10 Å. To fully describe the orientation of the protein with respect to the membrane, a second angle is required. We used the roll angle, which is shown and defined in Appendix A.

### 3.4. Changing Representation Scales

We have performed Monte Carlo insertion experiments for two other representation scales: the reparametrized Martini 3 representation described in Section 3.2 and a one-bead-per-amino acid representation of Levitt [29]. The latter scale is “trivial” in the sense that, since there is only one bead for each amino acid, the transfer energy of each bead corresponds a priori exactly to the experimental reference value for that amino acid. For this experiment, we used maltoporin (PDB code 1AF6) from [43]. Figure 10 shows the results, highlighting the distribution of transfer energies for the three levels of representation (A–C) and the forces acting on the inserted system in the Martini representation (D). It provides a pictorial representation of the distribution of transfer energy with a color code for each bead. This image gives a visual idea of the subtle differences between the parameter sets.

The resulting insertion depths and insertion angles for all three representation scales are given in Figure 11.

Figure 11 illustrates the effects of changing the representation scale for the insertion tests. Coarsening the representation smooths the surface of the potential energy, as seen in the shapes of the basins sampled for insertion depth and angle. Interestingly, the angle values are relatively robust to these changes. For insertion depth, the weighting shifts from a basin at −6 Å for the coarsest representation to a basin near −10 Å, although overall the same range of values is explored. It should be noted that the difficulty of sampling increases with the more accurate representations, so the exploration may not fully converge due to a potentially large number of local minima.

### 3.5. Application to a Double Membrane System: Membrane Fusion via the SNARE Complex

Here we look at a protein complex involved in exocytosis, in which molecules stored in lipid vesicles are transported from inside a cell to its surroundings. The final step of this process requires fusion of the vesicles to the plasma membrane and is mediated by SNARE (soluble N-ethylmaleimide sensitive factor attachment protein receptor) fusion proteins. We consider a model consisting of a four-helical SNARE bundle [44], which we have previously studied in interactive molecular dynamics simulations [45]. This complex comprises two transmembrane domains (TMDs) that anchor into the two membranes to be fused. We are pursuing the insertion of such a model, represented at the level of an elastic network model, using our BioSpring software, as described in [35]. This elastic network model provides flexibility to the protein and is described in the methods section.

We previously generated reference data from coarse-grained molecular dynamics simulations that accurately described the insertion of the isolated transmembrane domains, i.e., the transmembrane helices truncated from the core of the four-helix bundle [4]. Here, we investigate whether this precise membrane alignment also occurs in the overall complex when each of the two TMDs is inserted into their respective double membranes in a double-membrane arrangement in which the SNARE bundle connects both membranes, while the distance between them is changed interactively by the user. Such an experiment mimicking the membrane fusion process is shown below in Figure 12.

The first aspect to be examined is the resilience of the membrane insertion. As might be expected, the two TMDs are firmly seated in their respective membranes. Even when the membranes are moved closer together or apart, the interaction with the implicit membranes is strong enough to cause deformation of the elastic network model. This first experiment is not accurate at small distances between the two membranes because the membranes are likely to adopt a hemifusion state, which is not represented in this simple experiment. The most critical part in these interactive experiments is the accurate placement of each TMD in the membrane, which can be monitored using real-time analytical plots showing the average insertion depth for each TMD residue versus amino acid sequence. Such plots are commonly used in EPR water accessibility measurements on spin-labeled membrane proteins [46]. They can be measured experimentally. Here we note that the TMD insertion of the SNARE bundle at an intermediate state actually undergoes exactly the same insertion observed in the reference MD simulation. This feature is indicated by an arrow in Figure 12. This feature is preserved regardless of the scale of the representation, at Martini-level as well as at Levitt-level. The match is very close and occurs at an intermediate distance between the two membranes, commonly referred to in the literature as the SNARE docking state that usually precedes hemifusion. This membrane insertion was also validated against experimental EPR measurements as described in [4]. These results validate the setup and open the possibility for extended experiments where one may partially unzip the SNARE bundle to determine whether specific EPR signatures can be predicted for half-zippered complexes that have been postulated in the literature.

### 3.6. Application to a Curved Viral Membrane of Arbitrary Shape

We performed a preliminary simulation for the membrane insertion of a dodecameric AlphaFold2 model of the MetY domain from the hepatitis E virus replication polyprotein, as described in [47]. The construction of this dodecameric ring, which likely represents a membrane pore, was inspired by a general putative model of the membrane environment and in particular by a curved membrane shape that this protein might require for its function. The model, designed by Thibault Tubiana in the 3D modeling software Blender, is shown in Figure 13A. This is a hypothetical model of a cellular endomembrane, which is typically curved or tubular and can have a variety of shapes depending on the cell type in which it occurs. They can form complex structures such as vesicles or vacuoles. The hepatitis E virus (HEV) MetY protein is thought to be involved in the formation of a dodecameric pore that could potentially interact with such a cellular endomembrane. This interaction would allow regulation of what enters and what leaves the cell and provide structural support to certain parts of the cell. Because the mesh for such a large model represents a large number of mesh points and would require significant performance optimization of our software, we focused for now on an empirical model limited to the curved portion of the membrane near the insertion site of the protein and the corresponding putative pore, as shown in Figure 13B.

Figure 13C recapitulates the experiment we performed in which the protein assembly was successfully inserted and stabilized in the indicated membrane shape. The small bump at about 105 Å depth shows that when the protein enters into close contact with the membrane, its size causes it to tilt to one side, briefly increasing the IMP energy, overcoming a bottleneck and then pulling it back to the most stable position at ca. 104 Å, before overshooting and experience a steep rise in insertion energy.

These experiments with putative curved membranes are useful because insertion into a planar membrane in this particular dodecameric form of the MetY protein model does not yield results consistent with experimental observations. Therefore, a working hypothesis is that dodecamers can only adhere to such curved membrane pores with the correct size. Determining this optimal size is precisely the goal of follow-up studies with UNILIPID. Here we have shown the first proof of concept with a plausible pore model.

## 4. Discussion

Our implementation of the UNILIPID model is directly related to the original IMPALA method developed by Brasseur’s group. We have shown that UNILIPID corrects for limitations such as the inhomogeneous reproduction of reference transfer energies. By providing extensions to the model, we have made it more accurate and tunable. The novelty lies in our ability to use interactive software with improved calibration to position membrane proteins in implicit membranes of arbitrary shape. Our approach encompasses dual and even more complex membrane shapes found in membrane fusion applications. The concept of implicit membranes is not new, but the ability to use interactive software with improved calibration for complex membrane shapes in real time is. This opens the possibility of providing real-time analysis feedback and interactive refinement, as we have demonstrated for the SNARE system with insertion plots commonly used in EPR studies.

We provide a general framework that adapts to different scales of model representation, from very coarse models with one bead per amino acid, to the widely used coarse-grained Martini representations, to the most detailed models with all atoms. This feature is combined with a flexible representation of the implicit membrane, first by introducing an analytical implementation of a double membrane and then by generalizing to arbitrary shapes and membranes using an explicit mesh representation that serves as a reference plane for the original formula. The choice of a simple implicit approach is both an advantage, e.g., in terms of interactivity and simplicity of parameterization and fitting, and a limitation, e.g., in terms of phenomena such as lateral pressure or curvature stress [48] or the energetic details we include in the model, compared to more refined implicit representations as in [49]. Specific protein-lipid interactions are not easily modeled in this way. Rather, the goal would be to orient the membrane proteins prior to simulation or to test their presumed interactions with a range of membrane shapes.

Here we have focused in particular on the description of these new concepts and an initial reference implementation that we have tested on specific cases such as the OmpA membrane protein, maltoporin, the SNARE complex, and the MetY protein ring. In terms of current limitations, further performance improvements may be needed for efficient use, especially for large mesh models and macromolecular representations with a large number of beads. A critical step is the neighborhood search to calculate the normal for each particle with respect to the closest mesh surface grid point. We can use optimized methods such as the Neighbor Search Grid Burst (NSGB) package implemented in the UnityMol software by Xavier Martinez. We compared the UNILIPID implementation with reference simulations and validated the accuracy of the results. So far, we have performed a comprehensive comparison with one hydrophobicity scale and shown that our proposed method improves the parameterization by reproducing this scale exactly. In future work, an extended comparison of our results with actual partitioning data would be valuable. Although the present work focuses on improving the methodology by building on Brasseur’s original approach and characterizing the extension by mesh representations to handle arbitrary shapes, we can now move on to determining the best parameterization scale. In other words, the next step and question is: what is the best available hydrophobicity scale that can be used for this purpose, and what is the best associated representation scale. This is an investigation for a future paper. After agreeing on a preferred scale, after comparing many of the existing scales, we could compare our method with reference methods such as the OPM (Orientation of Proteins in Membranes) database in a proteome-wide study.

In UNILIPID, we chose an energy-based approach to characterize the lipid-protein interaction for membrane insertion, in contrast to e.g., a rule-based approach [50]. This choice makes biological sense as these interactions are widely characterized experimentally [51,52]. Computational approaches are abundant for such characterization, as for instance discussed in [53,54]. When computational power is not a limitation, molecular dynamics type approaches in fully hydrated explicit membrane environments may represent the gold standard [55]. Hence, there are many other methods of inserting proteins into membranes, including some approaches capable of treating curved membranes. One of the most widely used ones with this capacity may be the approach of positioning proteins in membranes (PPM) [56]. Furthermore databases such as MemProtMD provide insertion information for much of the structurally characterized membrane proteome [57]. These methods span several scales of representation. Our method emphasises computer interactivity to allow efficient interaction by a human operator and to provide intuitive feedback on the insertion properties of the membrane protein under study. To our knowledge, there is no other generally applicable interactive method for this purpose. In certain specific cases, such interactive simulations have been explored, e.g., for the extracellular matrix [58]. We consider them extremely valuable for hypothesis generation and exploratory evaluation of a system at the beginning of a new project. Another focus of our approach is to require little computational power and provide results quickly, typically within minutes. Another key feature of our method is its generalizability, providing the ability to handle different forms of membranes and a range of representations for the membrane proteins to be inserted. We used an example of a human-designed arbitrary membrane in the MetY example, yet alternatively such information could be obtained from experimental investigations, by segmenting 3D images for instance [36,37].

In the present study, we have shown how to dynamically vary the distance between two adjacent bilayers to mimic membrane fusion conditions. There are other parameters that can be dynamically varied. In [5], the actual thickness of the bilayer in the implicit IMPALA membrane was varied in discrete steps to estimate the optimal bilayer thickness in terms of the lowest insertion energy for a given membrane protein. These predictions were then compared to theoretical or experimental thickness data. With our tool, such an assessment can now be performed with a continuous variation of these parameters, for example by the user moving an appropriate slider while the interactive simulation is running.

To help interested readers experiment with the interactive approaches, we provide initial step-by-step instructions in the Appendix A. In it, the user is guided to start the interactive simulation and perform the first manipulations. We plan to add more videos of this kind in the future to describe all the functionalities implemented for these experiments. In the Appendix A, we demonstrate that these approaches can be used in a virtual reality context, although we do not have as many features available yet. Their implementation will be the goal of future extensions.

With such new tools in hand, one direction for future work is to explore functionally important membrane dynamics. This goal could be achieved relatively easily, for instance by using two planar sheets representing two membranes with different spacing to model fusion precursors found in systems similar to the SNAREs, for instance mitofusins or viral fusion models. As we have shown, the membrane sheets can be programmatically moved closer together or farther apart to mimic the biological process and investigate key intermediate steps in membrane fusion. Another exciting possibility is exploring the potential for advanced multiscale modeling based on membrane shape. A multiscale setup for this model could allow for the inclusion of diffusion within the membrane plane, as Wong has already demonstrated with a coarser shape-based rather than atom-based model to represent an extracellular matrix patch [58,59]. With UNILIPID, such a membrane embedding could be achieved at the bead level down to atomic precision and combined with lattice Boltzmann approaches, as we have explored in our work with Dry Opep [60]. This prospect opens up the possibility for hybrid mixed-resolution systems where an explicit lipid layer could be used to mimic specific interactions around a protein of interest, possibly with a coarse-grained lipid belt surrounding it, similar to what we did with water in our work on the WAT4 model [61]. The implicit membrane would then extend the membrane embedding to infinity on yet a different scale. Another possibility is to add more features to the UNILIPID model to account for the inherent dynamics of the membrane, e.g., by introducing deformations and undulations in the mesh representing the membrane. This improvement could be achieved by treating the membrane as an undulating and moving point cloud. Another idea to increase the sophistication of the model is to extend the treatment of hydrophobicity in these calculations to universal and unbiased methods, such as molecular hydrophobicity potentials (MHP) [62] or atomic solvation parameters (ASP) [63]. Another direction related to mixed membranes is the use of lipid-specific free transfer energy parameterizations to represent different membranes or even to mimic mixed compositions. This refinement could be done, for example, by randomly assigning free transfer energies proportional to the percentage of a given parameterized lipid type. In this case, an equivalent set of interactions, chosen at random, is scaled proportionally to the assigned lipid type. The major bottleneck is to obtain reference parameters for transfer energy in the form of individualized hydrophobicity scales for different phospholipids. We are considering such an extension for the future, possibly using purely simulation-based reference energies for parameterization. Association processes in the membrane are another target for improvement that could be achieved by adding attractive non-bonded terms or potentials related to the hydrophobic effect. Ultimately, it would be interesting to combine all these extensions, leading to a model with membrane diffusion and association processes that accounts for membrane undulations and their complex compositions with protein-protein stickiness and interactions. Such a model could provide insights into constrained diffusion, for example, along membrane layers in an organism.

## Figures and Tables

**Figure 1 membranes-13-00362-f001:**
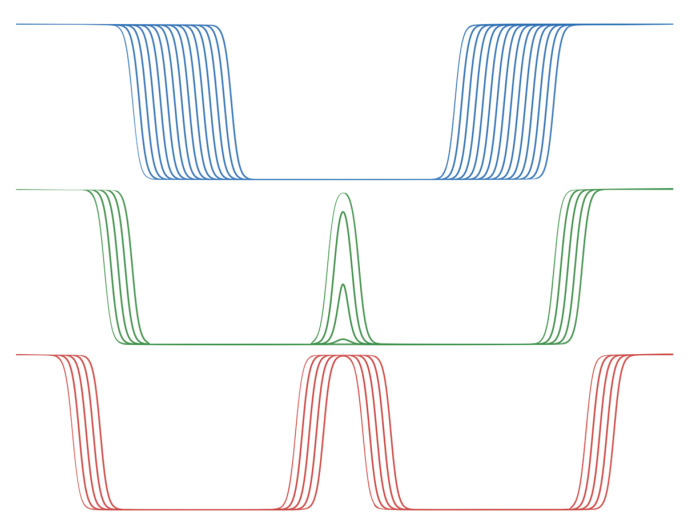
Functional form of C_double_(z) illustrating the membrane profile for three regimes of m. High values schematize the aqueous solvent, low values the membrane core. In blue, profiles ∀ m ∈ [0, 13.5]. In green, profiles ∀ m ∈ [13.5, 18]. In red, profiles ∀ m > 18.

**Figure 2 membranes-13-00362-f002:**
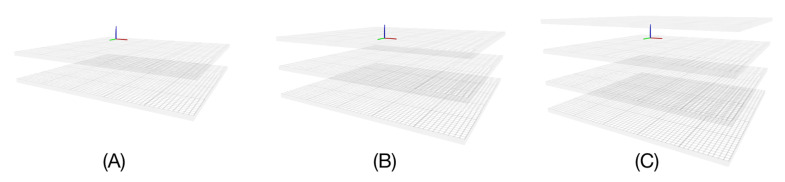
Representation of the double membrane under UnityMol. Combination of 4 cubic primitives structured in the form of a lattice whose transparency A varies for the “inner layers”, i.e., for the interfaces that merge from A = 0: completely transparent to A = 255: completely opaque (**A**) for m = 0 ≡ A = 0, (**B**) for m ≃ 15.75 ≡ A ≃ 255/2, (**C**) for m ≃ 30 ≡ A = 255.

**Figure 3 membranes-13-00362-f003:**
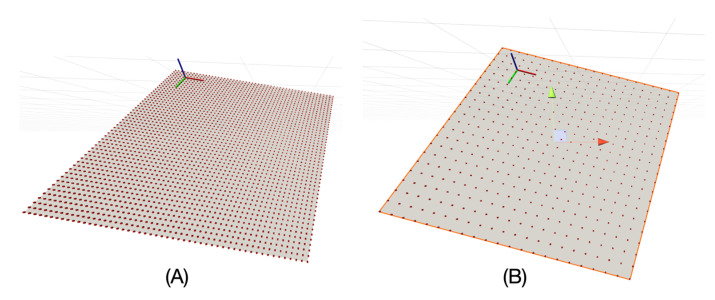
Plot of the reference surface in UnityMol showing the apparent vertices (red dots): (**A**) for a resolution of 4 Å; (**B**) for a resolution of 10 Å.

**Figure 4 membranes-13-00362-f004:**
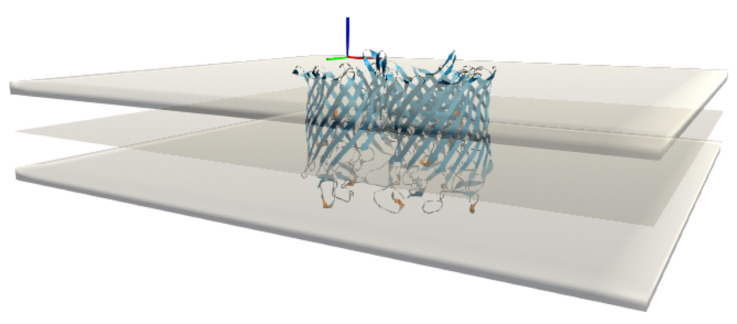
Illustration of maltoporin within a flat membrane reference mesh representing the centre of the bilayer, with both headgroup regions shown as thickened slabs in UnityMol.

**Figure 5 membranes-13-00362-f005:**
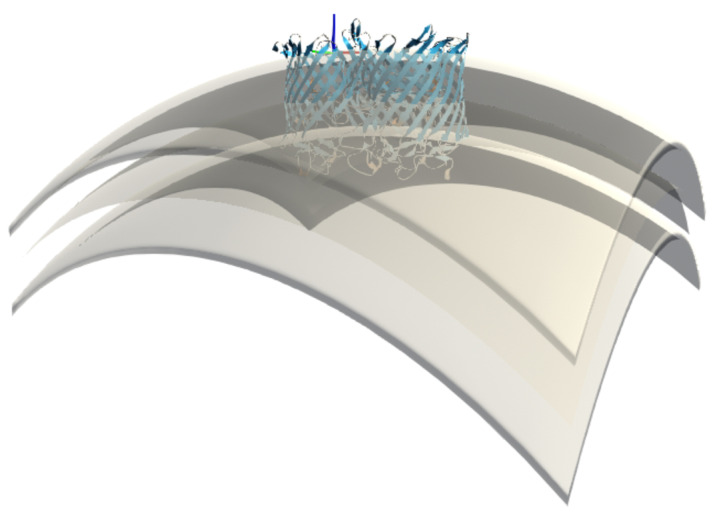
Illustration of maltoporin within a curved membrane reference mesh representing the centre of the bilayer, with both headgroup regions shown as thickened slabs in UnityMol.

**Figure 6 membranes-13-00362-f006:**
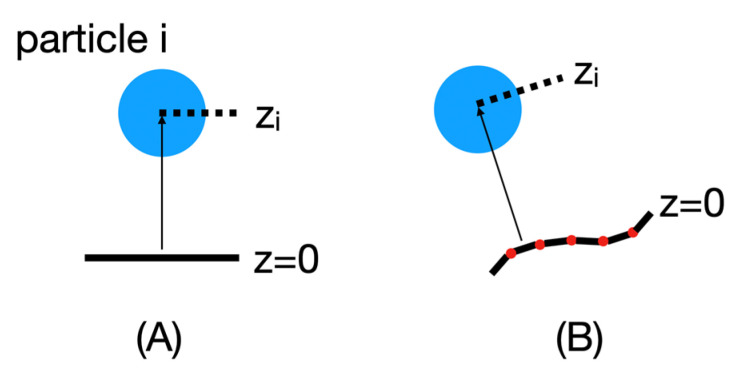
Schematic representation of particle position in the original planar implicit membrane frame (**A**) and in a mesh-based membrane of arbitrary shape (**B**). Particle i, represented by the blue circle, has position z_i_, which is used to evaluate the function C(z). Originally, z_i_ is simply the z-coordinate of the particle minus the z-coordinate of the membrane centre. In a mesh-based implementation, z_i_ becomes the distance of particle i from the membrane in the direction of the membrane normal.

**Figure 7 membranes-13-00362-f007:**
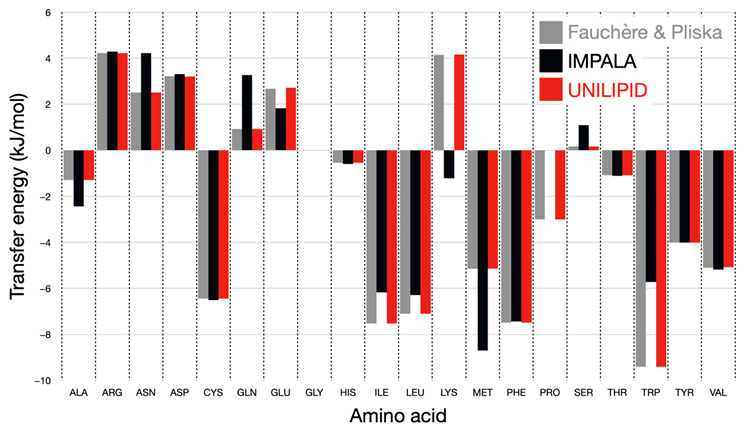
Bar graph of the transfer energy for the movement of each amino acid side chain into a membrane environment. The experimental reference scale is shown in gray, the calculated energies using the original IMPALA parameters in black, and the new UNILIPID parameterization in red.

**Figure 8 membranes-13-00362-f008:**
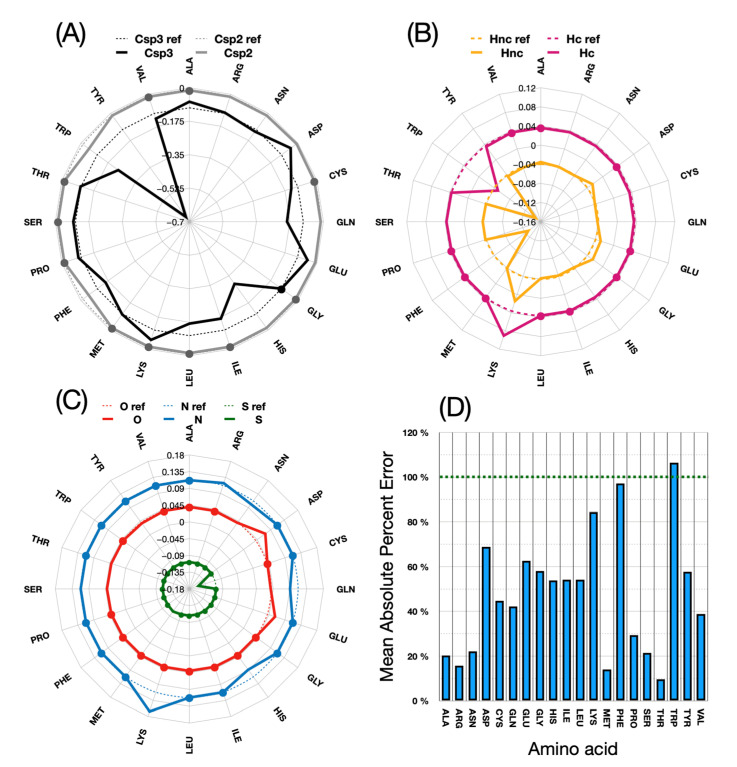
Comparison of the new UNILIPID energies with those of the reference implementation in IMPALA. (**A**–**C**) show values for the 7 atom types Csp3, Csp2, Hnc: H not charged, Hc: H charged, O, N, S connected by solid lines. The corresponding reference values are also plotted in the background, connected by dotted lines. Values that are identical to the original parameterization are indicated by dots. (**D**) Bar graph of the require parameter adjustment using the formula for mean arctangent absolute percent error (MAAPE): $\frac{1}{n}\overset{n}{\underset{i=1}{\sum }}\tan^{-1}\left|\frac{U_{i}-I_{i}}{U_{i}}\right|\times 100$ with *n*: number of UNILIPID (adjusted) parameters of the amino acid, *U*: *i*-th UNILIPID parameter, *I*: *i*-th IMPALA parameter.

**Figure 9 membranes-13-00362-f009:**
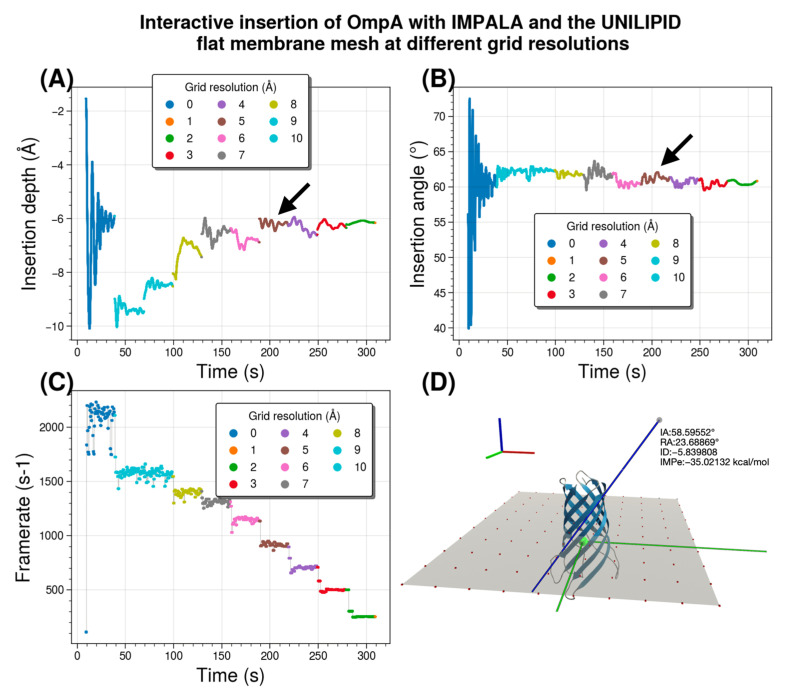
Comparison of an analytical membrane representation, given in the legend as “REF”, with a mesh representation with varying lattice spacing and decreasing lattice roughness through a sequence from 10 Å to 1 Å by 1 Å steps. Results are for the OmpA membrane protein. The time axis on the abscissa of figure panels (**A**–**C**) is the actual measured time. (**A**) Measured insertion depth for the series of experiments. A black arrow indicates a result close to the reference value. (**B**) Measurements of the insertion angle. A black arrow indicates values approaching the reference angle. (**C**) Simulation update rate for all experiments. (**D**) Graphical representation of a planar mesh with 10 Å resolution and an inserted protein.

**Figure 10 membranes-13-00362-f010:**
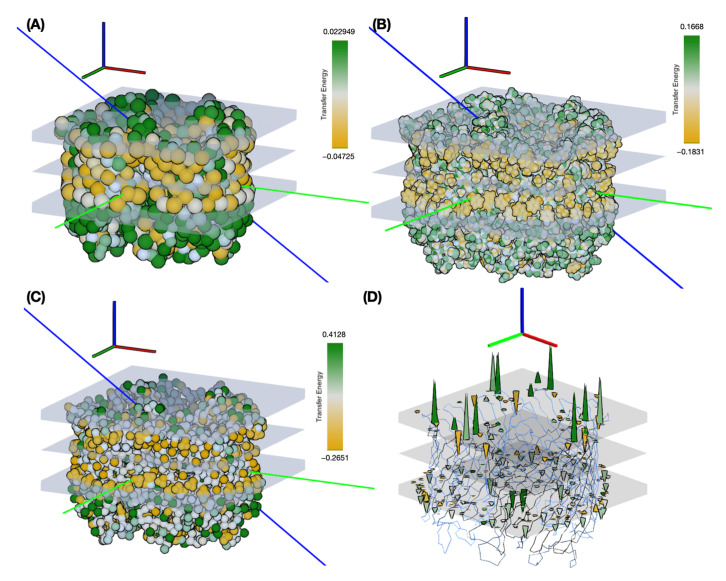
Interactive membrane insertion of maltoporin at different scales of representation. (**A**–**C**) Particles are colored according to their transfer energies, ranging from orange (lowest and negative energy if hydrophobic) to green (higher energy and positive if hydrophilic) to white for intermediate values. This color code was chosen to facilitate visual comparison with the original publication [19]. The implicit membrane is shown with the two interfacial regions and the z = 0 Å planes in gray. The blue vector is the insertion vector passing through the two particles T40 (CA) and Q48 (CA). (**A**) Levitt representation with one bead per residue. (**B**) All atom representation with calibrated UNILIPID transfer energy values fitted using the method described in Appendix A. (**C**) The Martini 3 representation shows the energies fitted to the octanol distribution values multiplied by the theoretical surface areas of the grains. (**D**) The same system and simulation setup as in (**C**), with the forces applied to the particles represented by colored spikes. The force amplitude is proportional to the length of each spike and the color represents the transfer energy of the corresponding particle. In fact, the forces are always directed orthogonally to the surface of the membrane on the plane of the interfaces.

**Figure 11 membranes-13-00362-f011:**
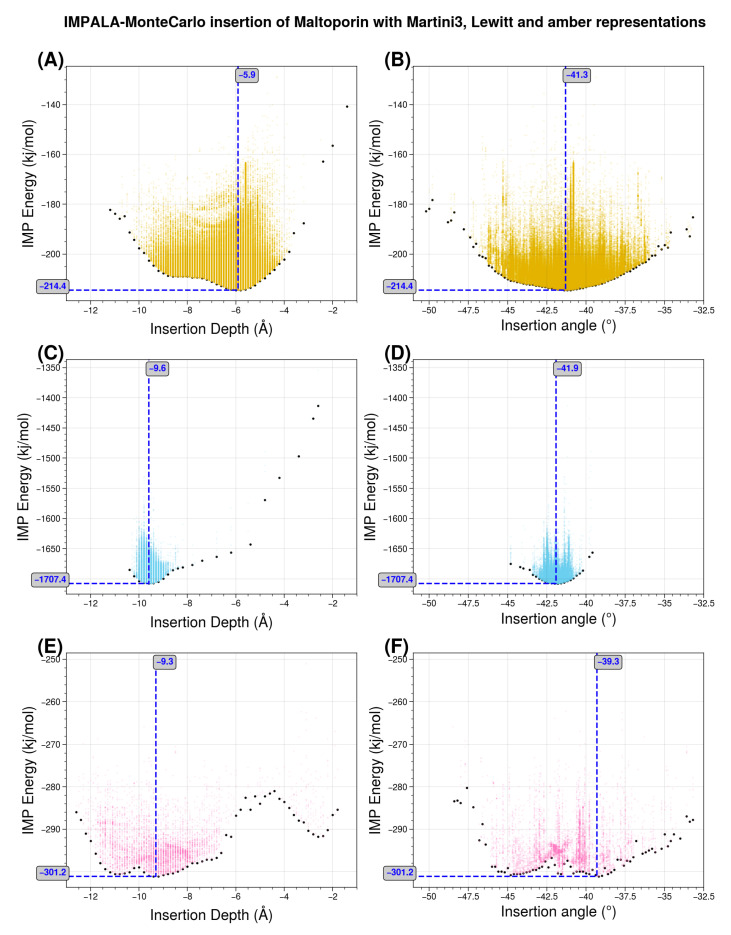
Monte Carlo insertion of maltoporin with three scales of representation: Levitt, Martini 3, and all-atom. The Monte Carlo parameters of temperature, translation, and angle are interactively adjusted to explore the space of angular and insertion depth values. The minimum energy depth and angle values are indicated by blue dotted lines and their values. (**A**,**B**) in orange the simulation using the coarsest model with the Levitt representation. (**C**,**D**) in blue, the results of the intermediate resolution Martini representation results with adjusted values for the transfer energy based on the octanol distribution and normalized to the theoretical surface area of the grains. (**E**,**F**) in pink, the most detailed Amber all-atom representation using the original values of [26] for comparison. The Martini 3 representation is closest to the original simulation reported by Brasseur’s team.

**Figure 12 membranes-13-00362-f012:**
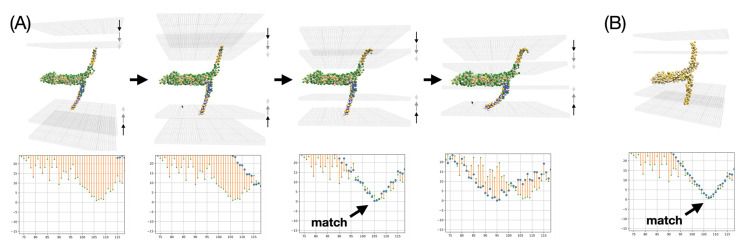
Experiments on the insertion of the SNARE system in double membranes. The four-helical SNARE bundle is represented in the implicit double membrane system delimited by grid lines at the position of the head group regions. The position of one of the TMDs is monitored by interactive representations of the insertion depth in terms of deviations from the reference values obtained by extended MD simulations [4]. The MD values are represented by small green dots, and the actual values in the UNILIPID simulation are represented by blue dots. Red lines highlight the differences between current values and reference MD data. (**A**) shows a series of four snapshots when the membranes are merged using the one bead per amino acid Levitt model representation. It shows good agreement with the reference data as highlighted by a thick black arrow in the third graph. (**B**) shows a similar result on a finer representation with a Martini 3 coarse grained representation. Again, very good agreement with the reference insertion pattern is observed.

**Figure 13 membranes-13-00362-f013:**
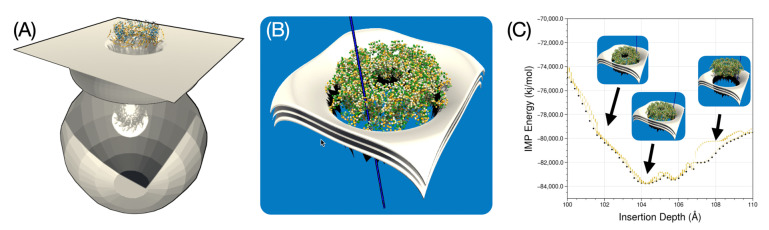
Experiments with curved membrane shapes to insert a dodecameric protein ring into such a bent membrane structure. (**A**) represents a thought model of the complex membrane shape to which the ring could be anchored. (**B**) is a close-range mesh we designed to test insertion into this membrane patch of arbitrary shape. The membrane midplane and head group regions are shown as layered sheets. (**C**) provides an overview of the insertion experiment and shows exploration of the insertion depth and three snapshots along the pathway. The approach to the protein ring is from the right side, representing the farthest distance, to a stable position (centre), and then to an exaggerated insertion (left) to illustrate how energy increases again in this last regime.

**Table 1 membranes-13-00362-t001:** Structural model parameters derived for several lipid types ^1^.

Lipid	DLPG	DOPC	DMPC	DLPE	DOPG	POPG	DPPC
Temperature	20	30	30	20	20	20	20
2D_C_	20.7	27.1	26.2	30.0	27.9	28.3	34.4
D_B_’	35.3 ^2^	35.9	36.9	42.1	42.8 ^2^	44.0 ^2^	47.8
β	7.3	4.4	5.4	6.1	7.5	7.9	6.7
z_0_	14.00	15.75	15.78	18.03	17.68	18.08	20.55
α	1.23	2.04	1.67	1.48	1.20	1.14	1.34
Ref.	[33]	[32]	[32]	[32]	[33]	[33]	[32]

^1^ Temperature is in °C; hydrocarbon thickness 2D_C_, steric thickness D_B_’, headgroup width β and z_0_ parameter are in Å; α is without a unit. ^2^ Thickness D_B_’ was extrapolated from reported thickness D_B_ by an 18% increment based on the average D_B_’/D_B_ ratio observed in Ref. [32].

## Data Availability

The software packages used for our implementation, UnityMol, MDDriver, and BioSpring, are open source projects, generally available via the websites https://sourceforge.net/projects/unitymol/files/ (accessed on 7 February 2023) and https://sourceforge.net/project/biospring/ (accessed on 7 February 2023). Our experiments were carried out with versions UnityMol_1.1.4_unilipid and BioSpring 1.0.1-5a92fad. The extensions to these software packages presented in this study are immediately available on request from the corresponding author and will become publicly available at the official sourcefourge sites after further testing by including them in the next software package release. To help experiment with our approach, we provide ready-to-use compiled versions for selected platforms along with specific input data. These data are openly available at https://recherche.data.gouv.fr/, accessed on 7 February 2023, at doi 10.57745/JRB0OF.

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
