# Peer review of "UNILIPID, a Methodology for Energetically Accurate Prediction of Protein Insertion into Implicit Membranes of Arbitrary Shape"

_membranes, 2023, doi:10.3390/membranes13030362_

Round 1

Reviewer 1 Report

This work presents an extended framework for the well known idea of the implicit membrane and proteins’ interactions with it, although there’s not much theoretical novelty in this approach. The account may be treated as a software description, but neither authors prepared their program for the user access. That’s why this generally interesting paper is a subject for major revision.

These and other points in detail:

1.       The title and general concept are blurry. Incorporation of what? Does the word “implicit” implies implicit membrane (spoiler: yes). What does the “framework” mean? Are you describing a theoretical method (probably not), certain results (I don’t think so) or a software (and we don’t have it)? Is it optimized for certain user features like real-time response, virtual reality experience or something else?

2.       What is the major application of UNILIPID? Insertion of the protein in the membranes, e.g. identification of the membrane binding modes? If so, it should be more clearly stated, starting from the title, abstract and introduction, since now it’s very vague what exactly do you propose. Moreover, if I’m right, where’s comparison with one of the most recognized analogous methods — Orientation of Proteins in Membranes (https://opm.phar.umich.edu), which exists as a proteome database and has a PDB integration? You should provide the performance and results comparison with methods like that if you introduce your own.

3.       Why should you rely on that weird seven types of atoms? Even despite you correct the previous errors of that “Impala”, why keep following it? There are more universal and unbiased methods to treat hydrophobicity in calculations, e.g. MHP (https://doi.org/10.2174/092986707779941050) or ASP (https://doi.org/10.1007/s002140000220). There’s too much mess in all that “atom type calibration methods 1–4”, which, in my opinion, should be cut from the paper for clarity and brevity. Moreover, you compare your current results for the amino acid transfer energy with that “Impala” and the Fauchere scale (Fig. 8), but why not comparing them with the experiments of the partitioning these compounds between water and non-polar phase? It’s crucial to see that your method provides good estimation of real results, not one of the many hydrophobicity scales.

4.       Critically lacks the comparison with other well-established methods. In 3.3, you study the “mesh parameters on the quality of results”. But what is that quality? As far as I see, it’s the correctness of the insertion of the membrane protein model inside the implicit membrane, and the reliability of these results. I think that a proteome-wide comparison with e.g. OPM is a good idea.

5.       What’s your novelty? I see that you don’t have any fundamentally new results, since implicit membrane is an old idea. Neither idea that you may design this membrane as an arbitrary 3D shape is new. What then? Interactive software to place the protein inside the complex membrane configurations iteratively? Or maybe VR applications (as I see from the youtube channel)? This is not highlighted in the paper, and therefore it’s hard to read and follow.

6.       Why “optimal” results (arrowed in Fig. 10) are for 5 Å, not 1 or 10? What’s the cause of such nonmonotonicity? Further, I don’t understand results in Fig. 10A and B, why there are separate torn data pieces? How is the insertion experiment conducted? Is it manual / interactive or it’s consistent scanning of depth/angle parameters, or some sort of MC trials? I didn’t find this in Methods, in contrast to 4 systems of atoms type and analytical description of double membrane. Moreover, just a single “angle” is not enough to completely describe the protein orientation with respect to the membrane, two are needed. But you don’t have them. Nor even depth and angle are illustrated in Fig. 10D.

7.       Where did “elastic network model” come from? It can be found in the SNARE’s results and Video S3 (and that’s amazing), but it’s not described in methods at all.

8.       What are results for SNARE? I caught only that you designed two tunable membrane layers and a responsive “elastic” protein, but what was your objective and the results? Without it, your text looks not like scientific paper but rather curious and unclear demonstration of unknown what :-(

9.       What’s the reason of that bizarre sphere / spherical segment of the membrane in Fig. 14? Looks curious, but what is it? Is there any sense to have this curved layer and not simple plane one? Do any results vary much? (I don’t think so.) What is the meaning of this curvature, or it’s just for fun?

10.    Finally: is this software available and what is it designed for? I tried to download a deposited archive from the recherche.data.gouv.fr and even launched that freaky UnityMol (which crushed perpetually), but it was completely unclear what to do next. By the way why chose such unusual software instead of standard PyMol? Beyond any doubt, you should prepare the software for users to try it, at least organize it in a sort of tutorial.

Author Response

Please see the attachment with a detailed point-by-point response for each of the 3 reviewers and a version of the manuscript with the tracked changes to highlight the extensive changes.

Reviewer 2 Report

UnityMol is available on SourceForge:

According to https://sourceforge.net/projects/unitymol/files/

several versions of UnitiMol binaries (Win, Mac) are available, the newest is UnityMol_1.1.4 2021-11-10

Which one was used by the authors?

I think such information should be updated in the manuscript.

BioSprong source is also available on SourceForge

/sourceforge.net/project/biospring/

Supplementary:

the link is broken and not working

https://youtu.be/B5Ro-712 azGOeY8,

even as:

https://youtu.be/B5Ro712azGOeY8

Author Response

(The authors gave the same response as above.)

Reviewer 3 Report

I find the work of Lanrezac and Baaden interesting since it is important to know how to reliably place proteins of different characteristics in biological membranes of different compositions and charges. However, I would like to make some comments on it.

1. The first paragraph on page 2 comments on the importance of the hydrophobicity of amino acids and their tendency to interact with hydrophobic systems such as the biological membrane and introduces the hydrophobic/interfacial scales. I think that at this point it would be very interesting to comment that the hydrophobicity scales do not take into account the secondary or tertiary structure of the proteins, which is a drawback in the methodology. If the global structure of a protein is known, wouldn't it be interesting to make a surface map of hydrophobicity which would help its insertion into the membrane?.

2. "Protein Positioning in Membranes" should be PPM and PPM should be "Protein Positioning in Membranes" (line 75).

3. Shouldn't some tools be commented on, such as those used in CHARMM-GUI or other ones ?.

4. Check lines 86-87 and 141.

5. Lines 236-237, "... open the possibility ... ". Would it be possible or not?. 

6. The phospholipids DLPG, DOPC, DMPC, DLPE, DOPG, POPG and DPPC have been used (Table 2). Would it be possible to use other lipids like PS, PI, SM, Chol, etc.?. Would it be possible to use other temperatures, higher/lower ?.

7. I have checked the Youtube videos as well as the data available at https://recherche.data.gouv.fr. I have not seen the possibility of using a membrane specifying the type or number of phospholipids, for example, simulating a plasmatic membrane. Can it be done or not or is it planned to be done in the future?.

Author Response

(The authors gave the same response as above.)

Round 2

Reviewer 1 Report

Now authors seem to have accounted for all my notes, thus the paper may proceed with the publication